# ADAM17: An Emerging Therapeutic Target for Lung Cancer

**DOI:** 10.3390/cancers11091218

**Published:** 2019-08-21

**Authors:** Mohamed I. Saad, Stefan Rose-John, Brendan J. Jenkins

**Affiliations:** 1Centre for Innate Immunity and Infectious Diseases, Hudson Institute of Medical Research, Clayton, Victoria 3168, Australia; 2Department of Molecular and Translational Sciences, Faculty of Medicine, Nursing and Health Sciences, Monash University, Clayton, Victoria 3168, Australia; 3Institute of Biochemistry, Christian-Albrechts-University, D-24098 Kiel, Germany

**Keywords:** lung cancer, tobacco smoking, KRAS, EGFR, ADAM17, IL-6 trans-signaling

## Abstract

Lung cancer is the leading cause of cancer-related mortality, which histologically is classified into small-cell lung cancer (SCLC) and non-small cell lung cancer (NSCLC). NSCLC accounts for approximately 85% of all lung cancer diagnoses, with the majority of patients presenting with lung adenocarcinoma (LAC). *KRAS* mutations are a major driver of LAC, and are closely related to cigarette smoking, unlike mutations in the epidermal growth factor receptor (EGFR) which arise in never-smokers. Although the past two decades have seen fundamental progress in the treatment and diagnosis of NSCLC, NSCLC still is predominantly diagnosed at an advanced stage when therapeutic interventions are mostly palliative. A disintegrin and metalloproteinase 17 (ADAM17), also known as tumour necrosis factor-α (TNFα)-converting enzyme (TACE), is responsible for the protease-driven shedding of more than 70 membrane-tethered cytokines, growth factors and cell surface receptors. Among these, the soluble interleukin-6 receptor (sIL-6R), which drives pro-inflammatory and pro-tumourigenic IL-6 trans-signaling, along with several EGFR family ligands, are the best characterised. This large repertoire of substrates processed by ADAM17 places it as a pivotal orchestrator of a myriad of physiological and pathological processes associated with the initiation and/or progression of cancer, such as cell proliferation, survival, regeneration, differentiation and inflammation. In this review, we discuss recent research implicating ADAM17 as a key player in the development of LAC, and highlight the potential of ADAM17 inhibition as a promising therapeutic strategy to tackle this deadly malignancy.

## 1. Introduction

Lung cancer is the primary culprit of cancer-related deaths among men and the second leading cause of cancer mortality among women worldwide, accounting for approximately 19% of all cancer deaths [1]. It is estimated that 1.8 million lung cancer cases are diagnosed each year worldwide [2]. Lung cancer is a heterogeneous disease which can be histologically classified into two major pathologies, small-cell lung cancer (SCLC) and non-small cell lung cancer (NSCLC), with the latter accounting for 85% of all lung cancer diagnoses. NSCLC is sub-classified histologically into lung adenocarcinoma (LAC), squamous-cell carcinoma and large-cell carcinoma, among which LAC is the most common form of NSCLC [3].

Although the past two decades have seen fundamental progress in the treatment and diagnosis of NSCLC, NSCLC is still often diagnosed at an advanced stage when therapeutic interventions are mostly palliative [4,5]. Typically, patients who are diagnosed with stage I, II and IIIA NSCLC undergo resection surgery to remove the tumour, followed by adjuvant therapy including radiation, chemotherapy and/or targeted therapy to prolong patient survival [4]. The first-line treatment for stage IV NSCLC patients is the non-selective chemotherapy cocktail of a platinum-based drug (Cisplatin or Carboplatin) with Paclitaxel, Gemcitabine, Docetaxel, Vinorelbine, Irinotecan, or Pemetrexed [4]. However, despite these strategies, the predicted 5-year survival rate for patients diagnosed with NSCLC is 15.9%, and has only slightly improved in recent decades [5,6]. Nonetheless, recent advances in genetic testing and biomarker discovery promise to foster the new age of drug development in the context of immunotherapies and personalised medicine to replace the old non-selective chemotherapy, and thus improve treatment outcomes [4].

## 2. Lung Cancer Risk Factors

### 2.1. Genetic

#### 2.1.1. *KRAS* Mutations

The family of *RAS* genes (*KRAS*, *HRAS*, and *NRAS*) are the first oncogenes identified, and are the most frequently mutated genes in human cancer cells, including lung, pancreatic and colorectal cancers [7]. In 1984, Santos et al., reported that mutations in *KRAS* are present in tumour tissues from lung carcinoma patients but not normal tissues [8]. Following this discovery, oncogenic *KRAS* mutations were identified as a common feature of LAC (accounting for 33% of all LAC patients) and other human cancers [3,9]. *KRAS* mutations are associated with tobacco smoking, and are critical for the initiation and the maintenance of LAC [10,11]. Experimental studies have shown that deletion of mutant Kras RNA resulted in apoptotic regression of both the early proliferative lesions and established lung tumours [10].

The *RAS* genes encode a family of membrane-bound guanosine triphosphate (GTP)-binding proteins that transduce signals between cell surface growth factor receptors and intracellular signaling pathways, and exist as binary molecular switches in two forms: GDP-bound (OFF or inactive) or GTP-bound (ON or active), both of which bind differentially to distinct intracellular effectors. The GDP–GTP conversion is tightly controlled by guanine nucleotide exchange factors (GEFs; which mediate GDP to GTP conversion) and GTPase-activating proteins (GAPs; which facilitate GTP to GDP conversion) [12,13]. RAS proteins share the same structure of the catalytic domain (i.e., G-domain), while they differ in their C-terminal hypervariable regions [13]. They undergo post-translational modifications including farnesylation, proteolytic cleavage of the C-terminus, carboxymethylation, ubiquitination, nitrosylation and palmitoylation, which are important in plasma membrane localization and interaction with their intracellular cooperators [13]. *RAS* mutations inhibit GAP-induced GTP hydrolysis (i.e., inactivation) of RAS proteins, resulting in their accumulation in an ON/active state [7].

RAS proteins regulate cell proliferation, differentiation, and apoptosis by interacting with signal transduction mediators, including rapidly accelerated fibrosarcoma (RAF), mitogen-activated protein kinase (MAPK), signal transducer and activator of transcription (STAT), phosphoinositide 3-kinase (PI3K), protein kinase C (PKC) and Ral guanine nucleotide dissociation stimulator (RalGDS) signaling cascades [14] (Figure 1). Moreover, oncogenic RAS proteins mediate amino acid uptake and synthesis, protein synthesis, as well as glucose uptake and metabolism [15,16]. Interestingly, analysis of 92 cell lines (including 64 *KRAS* mutant lines) from different disease settings using arrayed combinatorial small interfering RNA (siRNA) screens demonstrated that each cell line has a distinctive dependency on KRAS effectors (for example, RAF/MEK/ERK, PI3K/AKT or RAL effector pathways), with the majority of *KRAS* mutant cell lines being strongly dependent on either the RAF/MAPK pathway or the p90 ribosomal S6 kinases (RSKs) [17].

Many strategies have been developed over recent decades to target oncogenic RAS proteins, including targeting GDP/GTP binding and conversion, enhancing GTP hydrolysis/inactivation, inhibiting RAS post-translational modifications (i.e., farnesylation), and direct blocking of the downstream cooperators of RAS, such as RAF, MAPK, PI3K and RalGDS [13,18]. However, these therapeutic strategies have been unsuccessful due to many reasons; for instance, challenges in creating small molecules that are effective and selective to certain targets, the functional redundancy in RAS activation and its post-translational modifications, the activation of compensatory oncogenic pathways in response to RAS blockade, and the poor therapeutic index of such inhibitors [7]. As a consequence, tumours bearing *RAS* mutations are considered the most difficult to treat and are often excluded from treatment with targeted therapies [13]. However, to overcome this issue, recent studies have reported that preventing RAS-effector protein binding by developing small-molecule pan-RAS inhibitors or disrupting KRAS dimerization may represent therapeutic strategies to impair the oncogenic properties of KRAS, albeit yet unproven in preclinical cancer models [19,20]. Furthermore, the KRAS protein and DNA vaccines have emerged as a novel immunotherapeutic strategy to tackle oncogenic KRAS-addicted cancers in preclinical models or as adjuvant treatment options in clinical settings. For instance, the administration of such vaccines enhances antigen-specific cytotoxic T lymphocyte responses in vitro and in vivo, resulting in antitumor effects against KRAS-driven pancreatic and lung cancers [21,22,23]. The adoptive transfer of T-cells harbouring the engineered T-cell receptors (TCR) that specifically recognise mutated KRAS variants G12V and G12D has also been proven effective in reducing tumour growth in a *KRAS* mutant xenograft model [24].

In addition to attempts to directly target KRAS, another promising approach is to identify and inhibit novel downstream cooperators of RAS proteins without triggering the activation of other oncogenic mechanisms. In this regard, although the MEK/ERK pathway is not hyper-activated in RAS mutant human cancer cells, as indicated by steady-state levels of pMEK or pERK, MEK or ERK inhibition has been seen as an attractive way to treat RAS mutant cancers. However, in advanced NSCLC patients (incorporating KRAS mutant LAC), the systemic blockade of the ERK MAPK pathway with MEK inhibitors trametinib and selumetinib, used either as a monotherapy or in combination with standard chemotherapy (i.e., docetaxel), has yielded limited clinical benefit associated with acquired resistance and an unfavourable toxicity profile [25]. Furthermore, inhibition of SHP2 (also known as PTPN11), which links receptor tyrosine kinase signaling to the RAS–RAF–MEK–ERK pathway, in combination with MEK inhibitors has been shown to be effective in reducing tumour growth and preventing resistance to MEK inhibitors (i.e., trametinib) in preclinical models of cancer, including pancreatic cancer and NSCLC [26,27,28]. Furthermore, our group has recently shown that genetic and therapeutic targeting of the ADAM17 protease significantly abrogated LAC development via inhibiting ERK1/2 MAPK activation in LAC mouse models and xenografts [29]. Interestingly, ADAM17 inhibition did not lead to any compensatory over-activation of other oncogenic pathways, such as JAK/STAT and PI3K/AKT [29]. The role of the protease ADAM17 in cancer development will be discussed in detail in later sections of this review.

#### 2.1.2. *EGFR* Mutations

The epidermal growth factor receptor (EGFR) tyrosine kinase family was the first growth factor receptor family to be identified in cancer cells. It consists of four members, EGFR (ErbB1/Her1), Her2/neu (ErbB2), Her3 (ErbB3) and Her4 (ErbB4) [30]. Structurally, they are composed of an extracellular ligand-binding domain, a single membrane-spanning domain, a nuclear localization signal, and a cytoplasmic tyrosine kinase domain [31]. EGFR activation is regulated by the availability and the specificity of EGF family ligands, which are produced as cleavable cell surface precursor proteins and divided into three distinct groups, as follows. Firstly, EGF, transforming growth factor-α (TGF-α), epigen and amphiregulin bind exclusively to EGFR. Secondly, betacellulin, heparin-binding EGF (HB-EGF) and epiregulin bind to both EGFR and ErbB4. Thirdly, neuregulins (NRG1/2) bind ErbB3 and ErbB4, while NRG3/4 only bind to ErbB4 [30,31]. ErbB2 has no known ligand, and by heterodimerization, it assists in the ligand-activation of other family members [32]. For all EGFR family members with known ligands, ligand binding to the extracellular domains instigates conformational changes leading to receptor homodimerization or heterodimerization with other family members at the plasma membrane, allowing for the activation of the intrinsic receptor tyrosine kinase, which causes autophosphorylation of the cytoplasmic tails of the receptors [33]. The exception to this latter mechanism of activation is ErbB3, which signals only through heterodimerization with other family members as it lacks intrinsic kinase activity [34]. Upon ligand activation, ErbB4 is cleaved by membrane proteases, while its activated intracellular kinase domain translocates to the nucleus to control gene expression [35].

Collectively, ligand binding to EGFR family members regulates a plethora of cellular processes, including cell proliferation, survival, migration and angiogenesis, via activating a myriad of signaling pathways, for example, RAS/RAF/MEK/ERK, PI3K/AKT, Src tyrosine kinases, PKC, β-catenin and STATs [30,33]. It is, therefore, perhaps not surprising that dysregulated expression and/or activity of EGFR is observed in various epithelial cancers and is associated with poor patient survival. Such cancers include head and neck squamous-cell carcinoma (HNSCC) [36,37], NSCLC [38,39], colorectal cancer (CRC) [40,41], breast cancer [42], pancreatic cancer [43], ovarian cancer [44], gastric cancer [45], and brain cancer and glioblastomas [46,47]. With respect to lung cancer, *EGFR* mutations are present in 14% of all LAC patients, and are mutually exclusive with *KRAS* mutations [9]. Unlike *KRAS* mutations that are linked to smoking, those of *EGFR* predominantly arise in never-smokers, with a preponderance in female LAC patients from East Asia [11,48].

In 1993, Rusch and colleagues demonstrated that the protein expression of EGFR is significantly increased in 45% of tumour lesions from NSCLC patients, while TGFα was overexpressed in 61% of those tumours. These overexpression patterns were not detected in the adjacent normal tissues, suggesting that EGFR signaling is playing a role in developing NSCLC tumours [38]. This notion has paved the way for the development of orally-active EGFR tyrosine kinase inhibitors (TKIs), for example, Gefitinib (Iressa^®^), which binds the adenosine triphosphate (ATP) pocket in the EGFR catalytic domain without affecting insulin receptor tyrosine kinase activity, as a first-line targeted therapy for NSCLC [49,50]. In 2004, two landmark studies identified a causal role of *EGFR* mutations in NSCLC, where *EGFR* mutations correlated with patient responsiveness to Gefitinib therapy, and also, with growth inhibition by Gefitinib in NSCLC cell lines [39,51]. The role of those mutations was also confirmed in ‘never smokers’ patients with adenocarcinomas [52]. These *EGFR* mutations are heterozygous by nature and include small in-frame deletions and missense substitutions, which were shown to selectively activate PI3K/Akt and STAT signaling pathways, thus promoting cell survival and resisting cell apoptosis [53]. Moreover, *EGFR* mutations mediate the repositioning of critical residues juxtaposing the ATP-binding pocket of the tyrosine kinase domain of EGFR, resulting in stabilizing the interactions of the receptor with both ATP and TKIs [39,51].

Since EGFR signaling is implicated in the growth and progression of many cancers, a significant effort has been made to understand the full spectrum of its action in promoting cancer, thus informing the development of EGFR-targeted therapies. Monoclonal antibodies such as Cetuximab (Erbitux^®^) and Panitumumab (Vectibix^®^) have been developed to target the extracellular domain of EGFR or to block EGF and TGF α-induced activation of EGFR. These antibodies were FDA approved for use as monotherapy or in combination with other chemotherapies or radiation to treat HNSCC and CRC [54,55,56,57,58]. However, the use of monoclonal antibodies has been hampered by the development of antibody-dependent cell-mediated cytotoxicity [31]. EGFR TKIs were also developed as low-molecular-weight synthetic molecules that block the ATP binding pocket of the intracellular tyrosine kinase domain, which inhibits ligand-induced receptor autophosphorylation, leading to dampening of EGFR-dependent intracellular downstream signaling [31]. Several TKIs have been developed including Gefitinib (Iressa^®^), Erlotinib (Tarceva^®^), Lapatinib (Tykerb^®^) and Vandetanib (Caprelsa^®^) for the treatment of various malignancies, including HNSCC, NSCLC, breast cancer and CRC [31,59,60,61]. Moreover, the response of patients to TKIs (e.g., Gefitinib) is temporary with little to no longer-term clinical benefits due to tumours developing acquired drug resistance via secondary EGFR mutations or shifting their oncogenic dependence to EGFR-independent oncogenic pathways, including the KRAS/RAF/ERK pathway [31,62,63].

Taken together, these current limitations in targeting EGFR for long-term clinical benefit, plus the fact that effective therapies for NSCLC with mutation profiles typical for cigarette smoking are yet to be identified [31,62], highlight the urgent medical need to identify novel molecular targets to tackle NSCLC and other cancers.

### 2.2. Cigarette Smoking

Inhalation of environmental noxious particles has been proven to play a role in several lung diseases, including lung cancer and chronic obstructive pulmonary disease (COPD). Tobacco smoke, a well-known noxious agent and a pivotal risk factor for the development of lung cancer, is the leading preventable cause of mortality [64]. The association between lung cancer and cigarette smoking is unequivocal, with the trend of lung cancer incidence rates in a given country strongly aligned to its relative tobacco use [2]. However, the majority of smokers do not develop lung cancer, indicating that genetic, epigenetic and cellular factors including inflammation, oxidative stress, cellular senescence and injury may modulate the response of the lungs to noxious agents and determine the risk for developing lung neoplasia [64].

Exposure to tobacco smoke constituents instigates a cascade of events in the multistep process leading to pulmonary carcinogenesis [65]. Among the plethora of chemicals in cigarette smoke that are implicated in pulmonary carcinogenesis, the most carcinogenic compounds chemically belong to two groups: (1) polycyclic aromatic hydrocarbons and (2) nitrosamines. Moreover, the key ingredient, nitrosamine 4-(methylnitrosamino)-1-(3-pyridyl)-1-butanone, also known as nicotine-derived nitrosamine ketone (NNK), plays a crucial role in lung carcinogenesis [66]. NNK is a product of nicotine nitrosation during tobacco production and as a result of mammalian metabolism [67]. NNK levels in mainstream smoke (which originates from burning the cigarette) are 10–200 ng per cigarette, while its levels in sidestream smoke (exhaled by the smoker) are 50–100 ng per cigarette [68]. NNK is also implicated in thirdhand tobacco smoke, where it has been identified on the surfaces and in the dust of areas where smoking took place [69].

The carcinogenicity of NNK has been demonstrated in rats [70], Syrian golden hamsters [71], and mice [72,73]. In mice, NNK reproducibly induces bronchioalveolar hyperplasia, adenoma and adenocarcinoma in the lungs [73]. Moreover, NNK potentiates lung tumourigenesis in several genetically engineered mouse strains, such as the G-protein coupled receptor 5A (Gprc5a) knock-out mice [74], SPC/Myc transgenic mice overexpressing c-Myc [75], Clara (Club) cell protein CC10 knock-out mice [76], and the SPC/IgEGF transgenic mice overexpressing a soluble form of EGF [75].

At the molecular level, NNK metabolites are potent mutagens that induce activating mutations in the *Kras* proto-oncogene (due to the GC-to-AT transition at the second base of codon 12) and inactivating mutations in the *Trp53* tumour suppressor [73,77]. NNK induces the formation of the O-methylguanine adduct within ATII cells, resulting in *Kras* gene activation, followed by proliferation of ATII cells and malignant tumour formation [73]. Non-malignant human bronchial epithelial cells (HBECs) exposed to NNK in vitro also show malignant transformation with altered production of reactive oxygen species (ROS) and redox regulation [78]. NNK also induces cancer cell proliferation through engaging various signal transduction molecules and transcription factors, including ERK1/2 MAPK, p38 MAPK, PI3K/AKT, NF-󠇯κB, cyclooxygenase-2 (COX-2), B-cell lymphoma 2 (Bcl2) and c-Myc [79,80,81,82,83,84] (Figure 2). Despite the diversity of downstream effector molecules engaged by NNK, the full spectrum of molecular events which underpin its pulmonary oncogenicity remains unresolved.

### 2.3. Inflammation

There is accumulating evidence that the pathogenesis of lung cancer is associated with inflammation, particularly among cigarette smokers [85] (Table 1). A major risk factor for the development of lung cancer among smokers is COPD, which is manifested by persistent inflammation, oxidative stress, airway remodelling and injury of lung parenchyma [64]. The tumour immune microenvironment, comprising mainly of innate immune cells (macrophages and neutrophils), adaptive immune cells (B-cells and T-cells) and fibroblasts, has critical roles in dictating the characteristics of lung and other tumours, either by promoting chronic inflammation and tumourigenesis, or conversely delaying malignant progression via mounting of an anti-tumour immune response [86,87].

Poor prognosis in NSCLC has been associated with increased neutrophil accumulation in bronchoalveolar lavage fluid and in the alveolar lumen [88]. Alveolar macrophages have also been shown to play an important role in promoting lung carcinogenesis. Indeed, long-term depletion of macrophages markedly reduced tumour burden in a model of urethane-induced lung carcinogenesis [89]. Tumour-associated macrophages secrete pro-tumourigenic angiogenic factors, such as platelet-derived growth factor (PDGF) and vascular endothelial growth factor (VEGF), that fuel tumour cells and their growth [89,90]. *Kras* activation in Club (Clara) cells residing in the bronchiolar epithelium also induces a fulminant inflammatory response manifested by an abundant infiltration of alveolar macrophages and neutrophils [91]. In addition to these innate immune cells, adaptive immune IL-17-positive CD4 T cells and regulatory T cells are also associated with the development of lung cancer [92]. Recently, commensal microbiota were shown to instigate inflammation in LAC via activating lung-resident γδ T cells. In this scenario, local bacteria induce MyD88-dependent IL-1β and IL-23 secretion from myeloid cells, thus enhancing the proliferation and activation of γδ T cells, which induce IL-17-mediated inflammation and tumour cell proliferation. Eradication of these bacteria using germ-free mice or treatment with antibiotics markedly abrogated lung cancer development in a mouse model harbouring *Kras* mutation and *Trp53* deficiency [93].

Pro-inflammatory cytokines have also been implicated in the development of lung cancer (Table 1). IL-17 induces in vivo growth of NSCLC xenografts via promoting the net angiogenic activity and vascularity of tumour cells [94]. IL-6 can be secreted from *RAS* and *EGFR* mutant cells to promote tumourigenesis, an effect that has been attributed to IL-6 trans-signaling via the sIL-6R in *Kras* mutant LAC [95,96]. Indeed, the knockdown or blockade of IL-6 signaling using genetic approaches or neutralising antibodies abrogates *RAS* and *EGFR*-driven tumourigenesis via suppressing STAT3 activation [95,96,97]. Also, IL-8 expression is up-regulated in NSCLC cell lines and tumour biopsies, and corelates with impaired patient survival via promoting tumour angiogenesis [98,99,100]. Indeed, anti-IL-8 neutralising antibodies reduced tumour growth and metastasis in a xenograft model of NSCLC [101].

Inflammasome activation and subsequent production of IL-1β has also been shown to enhance tumour invasiveness, growth and metastasis [102]. Intriguingly, analysis of the Canakinumab Anti-inflammatory Thrombosis Outcomes Study (CANTOS), a randomised double-blind, placebo-controlled trial investigating the role of IL-1β inhibition in atherosclerosis using the selective anti-IL-1β monoclonal antibody Canakinumab (Ilaris^®^), revealed that lung cancer incidence and mortality were significantly lower in Canakinumab-treated patients compared to their placebo controls. This study alone suggests the potential role of IL-1β inhibition as a promising strategy for the treatment of lung cancer [103,104] (Table 1).

The ability of tumour cells to evade immune surveillance and facilitate tumour survival is due to their expression of surface molecules that interact with immune cells to maintain normal immune tolerance, which includes the interaction of the immune receptor programmed cell death 1 (PD1) with its tumour-associated ligand PD-L1 [6]. This interaction inhibits the anti-tumour functions of T cells including differentiation, proliferation and cytokine production [105]. Therefore, blockade of PD1/PD-L1 using monoclonal antibodies (e.g., Nivolumab (Opdivo^®^) and Pembrolizumab (Keytruda^®^)) provides an attractive way to restore the body’s immune system to attack tumour cells, which has been suggested as a treatment for NSCLC, particularly in PD1 high-expressing tumours [106,107,108,109]. However, the long-term clinical benefits from the use of such inhibitors are not clear [106,107,108,109]. In addition, PD-L1 has not been associated with the major driver mutations in NSCLC (*KRAS* and *EGFR*) [106,107,108,109], questioning the clinical relevance of using PD1/PD-L1 blockers in NSCLC.

## 3. The ADAM Family of Proteases

Many extracellular signaling proteins are synthesized as transmembrane proteins. The extracellular domain (ectodomain) of these proteins acts as its active form, which is shed from cells after cleavage by members of the ADAM family, which are structurally related, membrane-associated metalloproteinases [110,111] (Figure 3). The released active protein exerts its action mainly through paracrine signaling, and possibly via entering the bloodstream [112] (Figure 4). Proteolytic shedding of the ectodomain is an irreversible post-translational modification process, which controls the bioavailability of soluble signals and ligands that regulate cardinal processes including development, physiology, immunity, and pathology [110].

ADAM family members consist consecutively of an N-terminal signal sequence, a prodomain, a catalytic metalloprotease domain, a disintegrin domain, a cysteine-rich region, EGF-like domain, a transmembrane domain, and a cytoplasmic tail [112] (Figure 5). ADAM10 and ADAM17 are the best characterised ADAMs [110,112]. Although they share overlapping specificity for a repertoire of substrates, they have substrate preferences. This may result from the distinctive characteristics of their catalytic sites, the sensitivity to cellular signals and stimuli, the time required for processing of the substrate, the subcellular localization of the protease and its substrate and/or the cellular cues and contexts [111,113,114,115].

### 3.1. ADAM17

In 1997, two research groups reported ADAM17 as the TNFα converting enzyme (i.e., TACE) that releases bioactive TNFα from cells [116,117]. Since then, it has been shown that ADAM17 is responsible for the shedding of more than 70 membrane-tethered cytokines, growth factors, and cell surface receptors including sIL-6R, Notch receptor and EGFR ligands [118] (Figure 6). Therefore, ADAM17 is a pivotal switch for a myriad of physiological and pathological processes including cell proliferation, regeneration, differentiation, inflammation and cancer progression [118]. Moreover, ADAM17 is required for the normal development of various organs including the heart valves [119], eye, skin, hair, lungs [120,121] and mammary glands [122]. ADAM17 is ubiquitously expressed in human lung tissue [123], and its expression is up-regulated in lung diseases including asthma, COPD and endotoxin-induced acute lung injury [123,124,125].

Investigating the role of ADAM17 in health and disease in vivo has been challenging, since homozygous *Adam17* gene deletion results in perinatal lethality [120,126]. To overcome this, viable conditional ADAM17 knockout mice models that lack ADAM17 in certain compartments have been generated, for example, endothelial cells or leukocytes [127,128]. Moreover, radiation chimeric mice reconstituted with ADAM17^ΔZn/ΔZn^ hematopoietic cells, that lack functional ADAM17 due to targeted deletion of exon 11 harbouring its Zn^2+^-binding domain, have been used to demonstrate the pivotal role of ADAM17 in the shedding of leukocyte substrates [129]. Furthermore, a new strategy has been reported to generate viable mice with undetectable ADAM17 levels, which involved the creation of the *Adam17*ex allele via inserting a new exon into the *Adam17* gene, which started with an in-frame translational stop codon [130]. The new exon was flanked by altered canonical splice donor/acceptor sites, yielding a new artificial exon between exons 11 and 12, which when used, led to an abort of translation. Due to the non-canonical splice sites, the new exon was primarily used, leading to about 5% of ADAM17 expression compared to wild-type animals (Figure 7). The homozygous *Adam17*^ex/ex^ mice exhibited eye, skin and heart defects with reduced levels of soluble ADAM17 substrates [130]. The *Adam*^x/ex^ mice have been used extensively to identify the role of ADAM17 in inflammation and cancer [29,130,131,132].

### 3.2. Regulation of ADAM17 Activity

The sheddase activity of ADAM17 can be stimulated by a variety of stimulatory agents and signaling pathways, including phorbol ester, phorbol 12-myristate 13-acetate (PMA) [133], cytokines including TNFα, interferon γ (IFNγ) and IL-1β [113,134,135], Toll-like receptors (TLRs) [136], and G-protein coupled receptors (GPCR) [137]. Although ADAM17 activation has been studied extensively since it has been discovered, the molecular mechanisms by which cell signals stimulate ADAM17 have remained ambiguous. Indeed, how ADAM17 is activated is still puzzling molecular biologists, and the complexities of its regulation have been reported extensively previously [118,138,139], and for this reason, will only be briefly discussed below.

ADAM17 is synthesized as a catalytically inactive full-length precursor in the endoplasmic reticulum (ER). In the trans-Golgi network, this pro-form of ADAM17 undergoes a maturation step, which requires its inhibitory N-terminal prodomain—acting as a molecular chaperone for ADAM17—to be cleaved off by pro-protein convertase (furin protease), resulting in a catalytically active mature form of ADAM17 [140,141]. ADAM17 prodomain has been shown to interact and inhibit the catalytic domain of ADAM17, thus abrogating specifically ADAM17 surface activity [142]. Indeed, a stable form of the ADAM17 prodomain attenuated ADAM17-mediated disease models of sepsis, rheumatoid arthritis and inflammatory bowel disease via inhibiting TNFα secretion, thus providing in vivo evidence as a potential new highly selective inhibitor of ADAM17 [143]. Moreover, as will be discussed below, the ADAM17 prodomain can also mitigate LAC via selectively inhibiting sIL-6R shedding [29] (Figure 8).

The rhomboid protease family members (iRhoms), which are polytopic non-catalytic rhomboid-like intramembrane proteases, have emerged as critical regulators of constitutive and inducible shedding activity of ADAM17 [144,145,146,147,148]. Indeed, mouse embryonic fibroblasts (MEFs) lacking both iRhom1 and iRhom2 are devoid of ADAM17 shedding activity [144,146]. iRhom2 has been reported to mediate the trafficking of ADAM17 from the ER, which is suggested to be the rate-limiting step for ADAM17 maturation [145], and consequently, regulating ADAM17 shedding activity [144,149]. Here, the transmembrane domain of ADAM17 is suggested to be required for the binding of ADAM17 to iRhom2 [144]. Further evidence for a key role of iRhom2 in regulating ADAM17 is the observation that while functional ADAM17 has been detected in exosomes, inhibition of iRhom2 suppressed exosomal ADAM17 release [150] (Figure 8).

The phosphorylation of the cytoplasmic domain of ADAM17 also plays a pivotal role in controlling its sheddase activity. Specifically, phosphorylation of Threonine 735 residue (Thr^735^) by MAPKs, including ERK and p38 MAPK, can enhance ADAM17 shedding activity, and this ERK-mediated Thr^735^ phosphorylation can also induce ADAM17 trafficking to the cell membrane [151,152,153,154,155] [156]. Of note, cellular senescence is associated with ADAM17-mediated release of amphiregulin and TNF receptors from the surface of senescent cells via p38 MAPK-induced phosphorylation of ADAM17 cytoplasmic tail [157]. Interestingly, p38 (but not ERK) has been shown to phosphorylate (i.e., activate) ADAM17 on Thr^735^, thus enhancing its shedding of the pro-tumourigenic and pro-inflammatory sIL-6R in LAC models [29].

Another regulatory mechanism of ADAM17 activity involves tissue inhibitor of metalloproteinases-3 (TIMP3), which inhibits ADAM17 activity via interacting with its ectodomain [158,159]. Indeed, genetic ablation of TIMP3 enhances ADAM17 activity and TNFα shedding, leading to inflammation [160]. The activity of ADAM17 can also be regulated via dimerization involving its cytoplasmic domain (also required for its interaction with TIMP3), and ADAM17 can be present as dimers at the cell surface during steady states [154]. With respect to the above-mentioned regulation by MAPKs, upon activation by ERK or p38 MAPK signaling, ADAM17 dimers dissociate and ADAM17 is presented as monomers, allowing for the release of TIMP3 [154].

Other mechanisms of regulating ADAM17 activity include the disruption of the actin cytoskeleton, probably via a mechanism independent of p38 MAPK or ERK-induced ADAM17 phosphorylation [161]. In this scenario, the actin-binding protein, filamin, has been proposed to regulate ADAM17 activity since disrupting the ADAM17–filamin interaction inhibited ectodomain shedding of CD44 and amyloid protein precursor [162]. Additional mechanisms of ADAM17 regulation, albeit ill-defined, involve the tetraspanin CD9, redox modifications of the cysteinyl sulfhydryl groups in mature ADAM17 (i.e., exclusive of its prodomain and intracellular region), and PKCε [163,164,165]. ADAM17 activity could also be regulated during its transport to Golgi apparatus by sequestering in lipid rafts, since depletion of membrane cholesterol enhances ADAM17-dependent shedding of TNFα and TNF receptors [166].

Interestingly, Dang et al. has proposed that ADAM17 substrate selectivity may be modulated by certain signaling pathways in response to the inducers PMA and angiotensin II, irrespective of enhanced ADAM17 protease activity [167]. Activation of PKC-α and the PKC-regulated protein phosphatase 1 inhibitor 14D is required for ADAM17 cleavage of TGF-α, HB-EGF and amphiregulin, while PKC-δ is required for NRG release [167]. Moreover, a juxtamembrane segment in the extracellular domain of the ADAM17, regulated by a protein-disulfide isomerase, has been identified as a sensor that binds IL-6R, but not TNFα, and mediates its shedding [168]. These studies indicate that substrate selection by ADAM17 could be influenced in certain settings by different mechanisms.

## 4. ADAM17 and Its Role in Cancer

Since ADAM17 mediates the ectodomain shedding of various pro-tumourigenic cytokines, growth factors and surface receptors [138], it is of no surprise that ADAM17 has attracted attention as a potential driver of cancer. In support of this notion, ADAM17 is overactivated or overexpressed in numerous human cancers, including NSCLC (and its major LAC subtype), colon carcinoma, breast cancer, hepatocellular carcinoma, HNSCC and gastric cancer [29,169,170,171,172,173,174,175,176,177]. Furthermore, ADAM17 targeting using genetic, antibody-mediated or pharmacological methods suppresses cell proliferation and tumour growth in cancer models for LAC [29,131], CRC [132], breast cancer [178], prostate cancer [179], pancreatic cancer [180] and ovarian cancer [181]. Some of the therapeutic agents that were developed to inhibit ADAM17 are summarized in Table 2.

Despite the success of targeting ADAM17 using small-molecule inhibitors (SMIs) and siRNA in cancer models, the clinical benefits of such approaches are questionable. Indeed, the caveat of using SMIs (e.g., hydroxamates such as GW280264X) to inhibit ADAM17 is their non-specificity since they target other proteases (e.g., ADAM10) leading to off-target effects and unfavourable toxicity profiles that have resulted in their failure in the clinic [182,187,188]. For example, potent non-selective inhibitors of ADAM17 and ADAM10, such as INCB3619 and INCB7839 (aderbasib), have shown promise in reducing tumour growth alone or in combination with other chemotherapeutic agents when tested in preclinical models of breast cancer and NSCLC [182,183,184,189]. However, their efficacy in the clinic has been stunted due to safety and toxicity concerns likely due to the induction of deep vein thrombosis in some patients [190]. The inhibitor INCB7839 is currently under investigation in the clinical trial NCT02141451, in combination with Rituximab (a monoclonal antibody targeting CD20), for the treatment of Diffuse Large B Cell Non-Hodgkin Lymphoma.

In addition, using siRNA to inhibit ADAM17 expression is problematic to translate, and is considered a questionable approach since suppressing ADAM17 expression does not necessarily affect its activity [191,192]. Similarly, transgenic over-expression of ADAM17 in mice, including in the lung, does not result in increased shedding (i.e., protease) activity, thus indicating that ADAM17 activity is not dependent on its transcriptional regulation [193]. Therefore, there is a clear need to identify and develop new non-toxic and selective inhibitors of ADAM17. In this regard, the ADAM17 prodomain has been identified as a potent inhibitor of ADAM17 surface activity, which significantly reduced tumour burden in LAC models [29,131]. In addition, the recent advent of specific antibodies to inhibit ADAM17 activity provides another avenue to specifically target ADAM17 whilst minimizing any off-target effects [180].

### 4.1. ADAM17 and Its Substrates in Lung Cancer

While the role of ADAM17 and its specific substrates (e.g., EGFR ligands, Notch, sIL-6R) has been previously documented in several cancers (e.g., lung cancer, CRC, pancreatic cancer, breast cancer) [29,41,42,43,61,97,132,177,182,187,194,195,196,197], for the purpose of this review, we will focus on lung cancer, for which many mechanistic insights are relevant to other cancer types.

#### 4.1.1. EGFR Ligands

The role of EGFR signaling in the pathogenesis of cancer, including the lung, is undisputable [51,53,96,182,198,199,200]. EGFR ligands have the ability to stimulate their receptors in autocrine, paracrine and juxtacrine modes [112]. It has been suggested that the uncleaved membrane-tethered EGFR ligands hinder the dimerization of the ligand–receptor pairs. Therefore, shedding of the ligand removes this impediment, resulting in enhanced dimerization and EGFR signaling [112]. Surprisingly, overexpression of uncleavable forms of TGFα or HB-EGF could also enhance EGFR signaling, mainly through enhancing juxtacrine signaling [201,202,203], and also by preventing endocytosis and down-regulation of the receptor [204]. However, it has been suggested that the transmembrane form of the ligand could activate EGFR via juxtacrine signaling only after its shedding via ADAM17 [195]. Nonetheless, these data place ADAM17 at the centre of EGFR signaling.

In the context of lung cancer, amphiregulin has been shown to inhibit cell apoptosis in NSCLC cell lines via insulin-like growth factor-1 (IGF1)-dependent mechanism [205]. In addition, tobacco smoke exposure to NSCLC cell lines can enhance ADAM17 activation, culminating in the release of amphiregulin and EGFR activation [165]. Interestingly, serum levels of amphiregulin and TGFα predict poor response to Gefitinib in patients with advanced NSCLC [206].

#### 4.1.2. Notch Signaling

Aberrant Notch signaling has been implicated in the development of numerous human malignancies, including acute lymphoblastic leukemia, breast cancer, melanoma and LAC [196]. There are four different Notch receptors (Notch 1–4) and five ligands (Delta-like 1, 3, and 4 and Jagged 1 and 2) [196]. Notch signaling is tightly controlled via cleavage of Notch receptor at distinctive sites. Firstly, a furin-like convertase cleaves Notch receptor at the ectodomain (S1) site to create a cell surface receptor [207], which is resistant to proteases in the absence of Notch ligands [208]. Therefore, engaging the Notch ligand allows for the generation of a second proteolysis site (S2) to be processed by ADAM proteases. Subsequently, the aspartyl protease ϒ-secretase carries out cleavage at sites S3 and S4, producing Notch intracellular domain (NICD) and culminating in its nuclear translocation and activation of Notch target genes, including *HES1* and *HEY1* [209].

Although Notch1 is a substrate of both ADAM10 and ADAM17, it has been suggested that ADAM10 can only process the ligand-associated receptor, while ADAM17 cleavage activity is independent of Notch ligands [210]. In the context of NSCLC, Notch signaling exerts pro-oncogenic role via enhancing tumour cell survival under hypoxic conditions [211]. Furthermore, Notch1 deficiency or inhibition significantly inhibited *Kras*^G12D^-driven LAC [212]. Moreover, ADAM17 is shown to be required for the proteolysis of Notch1 receptor, which enhances EGFR expression and EGFR-dependent activation of ERK but not AKT in NSCLC cell lines in vitro [187].

#### 4.1.3. sIL-6R-Mediated IL-6 Trans-Signaling

Previous studies have mainly focused on EGFR family ligands and the Notch receptor as downstream processed substrates of ADAM17. Given that ADAM17 sheds over 70 substrates, there lies the need to investigate the potential role of other ADAM17 substrates in cancer. In this respect, IL-6 trans-signaling via sIL-6R has recently emerged as a central driver of *Kras* mutant LAC and CRC, the latter via EGFR signaling which paves the way to potentially exploit components of the IL-6 trans-signaling axis (including ADAM17) as therapeutic targets in patients who develop resistance to EGFR blockade [29,97,132,197].

Classical IL-6 signaling requires the binding of IL-6 to the IL-6R complex, which consists of the transmembrane proteins gp130 and IL-6R [213]. IL-6 classical signaling is thought to mediate the protective immune responses of IL-6 during host defense, homeostasis and tissue regeneration in steady-state conditions [213]. In contrast, IL-6 trans-signaling takes place when the soluble form of IL-6R binds to IL-6, following which, the IL-6/sIL-6R complex acts as an agonist by binding to the ubiquitously expressed gp130 (Figure 9). This confers IL-6 responsiveness to cells that do not express the IL-6R and augments the spectrum of effects of classical IL-6 signaling in cells expressing IL-6R [138,213,214]. IL-6 trans-signaling has been implicated in the pathogenesis of numerous inflammatory conditions and malignancies, the latter including those to which ADAM17 has also been associated with, namely CRC, pancreatic cancer and lung cancer [29,97,213,214,215,216,217].

Regarding lung cancer, blocking of IL-6 trans-signaling using sgp130Fc—which is a fusion protein of sgp130 and the crystallizable fragment of immunoglobulin G1—or the anti-IL-6R monoclonal antibodies 1F7 and 25F10, ameliorated tumourigenesis in the oncogenic *Kras*^G12D^-induced LAC mouse model [29,97,218]. Furthermore, in an LAC-associated cachexia mouse model driven by oncogenic *Kras*, IL-6 trans-signaling, via modulating STAT3, has been shown to promote muscle wasting [197].

ADAM17 and ADAM10 are the main sheddases of human and murine sIL-6R. While ADAM17 releases sIL-6R in an induced manner, ADAM10 is responsible for the constitutive release of sIL-6R, which can be compensated for by ADAM17 [219]. Notably, the predominant role of ADAM17 in mediating the pathological consequences of deregulated sIL-6R-driven IL-6 trans-signaling has recently been reported in *Kras* mutant LAC and cigarette smoke carcinogen (NNK)-induced lung cancer [29,131]. In this regard, ADAM17 is overactivated via p38 MAPK-mediated threonine phosphorylation, following which, activated ADAM17 drives the preferential processing of the sIL-6R, which induces lung cancer cell proliferation through ERK1/2 MAPK activation. Moreover, inhibition of ADAM17 using genetic and prodomain inhibitor approaches significantly impaired tumour growth in LAC models and xenografts [29,131]. Interestingly, the protection against LAC formation in the *Kras*^G12D^ model was greater upon blockade of ADAM17 versus IL-6 trans-signaling, which raises the tantalizing prospect that ADAM17 may also employ additional processes substrates, albeit in a minor role, to promote oncogenic Kras-driven LAC [29,97].

## 5. Conclusions and Future Remarks

In summary, the therapeutic targeting of ADAM17 represents an attractive strategy to tackle LAC and other conditions of the lung (e.g., asthma, emphysema), and for that matter, other cancers (e.g., CRC and pancreatic cancer) with aberrant ADAM17 activity and/or expression. Moreover, ADAM17 inhibition may serve as a new indirect avenue to target the oncogenic activities of mutant KRAS in cancers including lung cancer (i.e., LAC), CRC and pancreatic cancer. In the context of lung cancer, it will be of interest to assess the driver role of ADAM17 in other lung cancer subtypes (e.g., *KRAS* wild-type LAC, EGFR mutant LAC, squamous cell carcinoma). Furthermore, in light of the preferential requirement by ADAM17 for the IL-6R substrate to facilitate its pro-tumourigenic actions in *KRAS* mutant LAC, the prospect of targeting ADAM17 versus sIL-6R-dependent IL-6 trans-signaling in LAC is worth considering. Current strategies to target IL-6 in the clinic have relied upon antibody-mediated approaches against IL-6 or IL-6R that block both pathological trans-signaling and homeostatic (i.e., protective) classical signaling [213]. However, a major drawback with such antibody therapies in the clinic, such as tocilizumab and sarilumab, is that they invariably cause side effects comprising infections due to an impaired host defence against bacteria, imbalanced metabolism leading to higher blood cholesterol levels, and perforated lesions in the gastrointestinal tract [213,220]. Since the therapeutic targeting of ADAM17—with a highly specific and non-toxic prodomain inhibitor—in the preclinical *Kras*^G12D^ LAC model displayed enhanced anti-tumour activity compared to anti-IL-6R antibodies [32,100], the use of ADAM17 inhibitors to specifically block pathological IL-6 trans-signaling in the lung promises to be a more effective and safer strategy to ameliorate disease states, including KRAS mutant LAC, driven by IL-6 trans-signaling. In this regard, the identification of sIL-6R as a key molecular read-out for ADAM17 activity in LAC has potential for clinical translation, whereby sIL-6R could be exploited as a surrogate biomarker (i.e., released into the blood of patients for detection by ELISA) for ADAM17 activity to assist in the stratification of LAC patients predicted to respond to anti-ADAM17 therapies. Despite the promise of targeting the ADAM17-sIL-6R axis in LAC, we do nonetheless acknowledge the possible involvement, albeit minor, of additional ADAM17 substrates (among the >70 known) in LAC. Accordingly, future work is warranted to investigate the full substrate repertoire of ADAM17 in LAC, for instance, using mass-spectrometric-based proteomic analyses, including the terminal amine isotopic labelling of substrates (TAILS) method, to identify new drug targets and biomarkers, and to reveal the role of ADAM17 in other disease modalities.

## Figures and Tables

**Figure 1 cancers-11-01218-f001:**
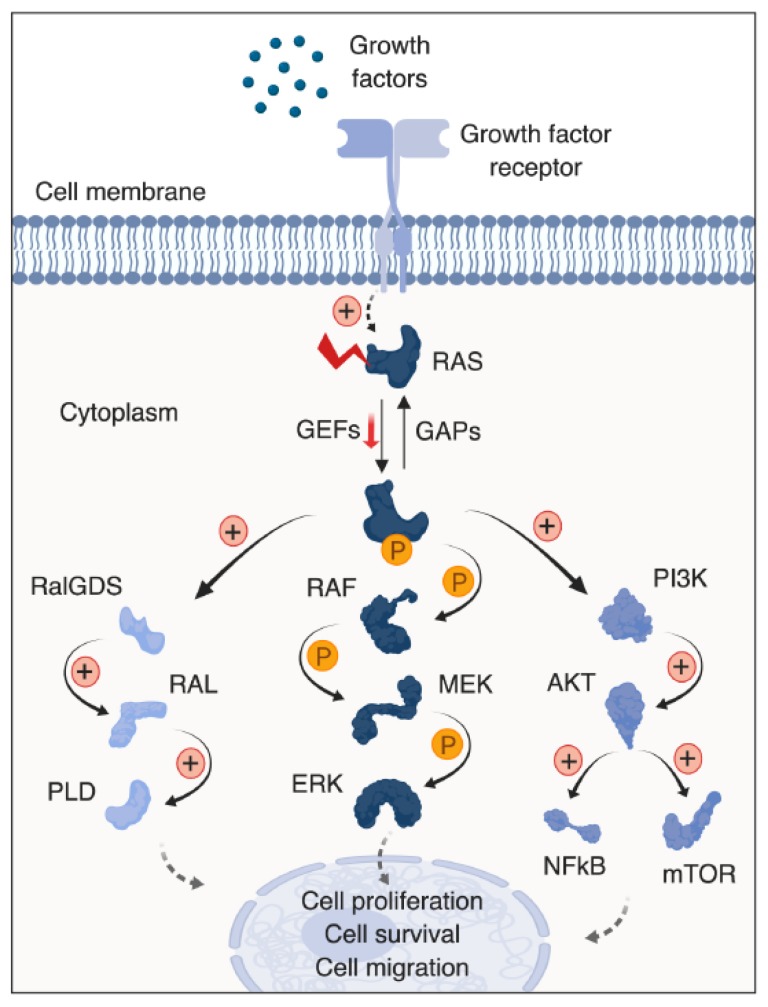
Signaling pathways engaged by RAS proteins. RAS mutations inhibit GAP-induced GTP hydrolysis, resulting in accumulation of active RAS. Abbreviations denote: GAPs; GTPase-activating proteins, GEFs; guanine nucleotide exchange factors, RalGDS; Ral guanine nucleotide dissociation stimulator, RAL; RAS-related protein, PLD; phospholipase D, RAF; rapidly accelerated fibrosarcoma, MEK; Mitogen-activated protein kinase kinase, ERK; extracellular signal-regulated kinase, PI3K; phosphoinositide 3-kinase, NF-κB; nuclear factor kappa-light-chain-enhancer of activated B cells and mTOR; mammalian target of rapamycin.

**Figure 2 cancers-11-01218-f002:**
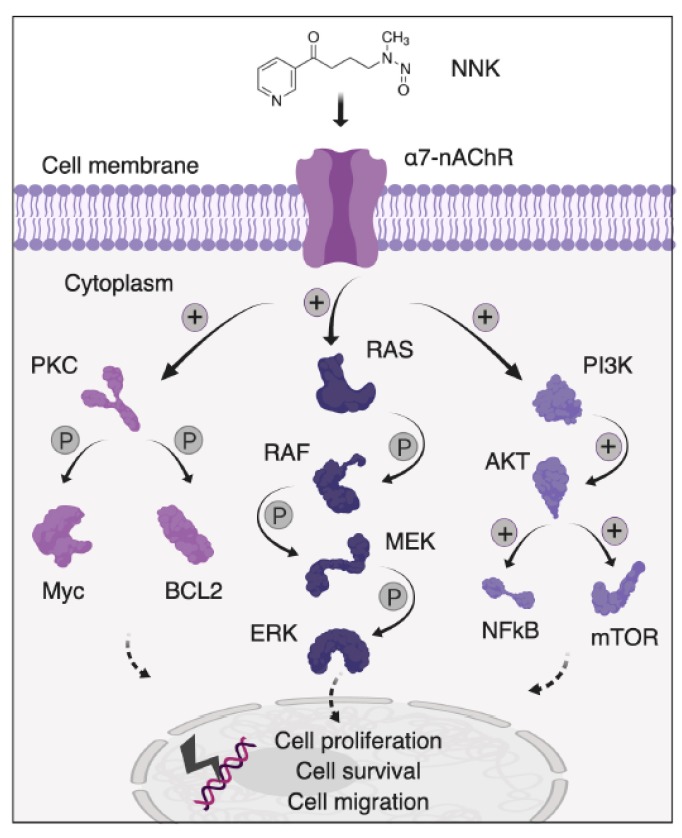
NNK-mediated activation of oncogenic signaling pathways. Abbreviations denote: NNK, nicotine-derived nitrosamine ketone; nAChR, nicotinic acetylcholine receptors; PKC, protein kinase C; BCL2, B-cell lymphoma-2; RAF, rapidly accelerated fibrosarcoma; MEK, mitogen-activated protein kinase kinase; ERK, extracellular signal-regulated kinase; PI3K, phosphoinositide 3-kinase; NF-κB, nuclear factor kappa-light-chain-enhancer of activated B cells; mTOR, mammalian target of rapamycin.

**Figure 3 cancers-11-01218-f003:**
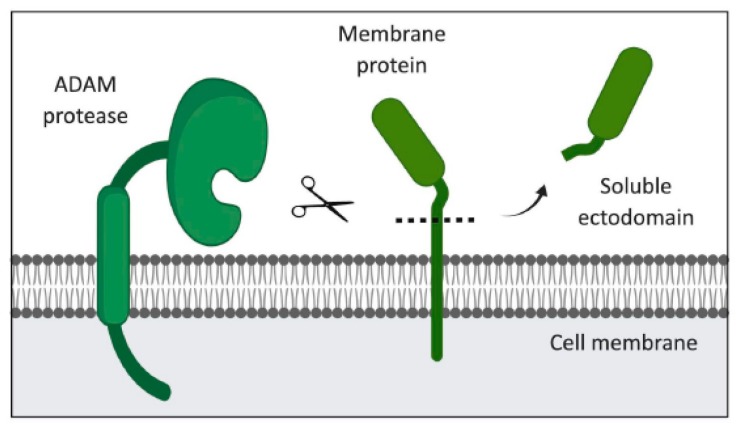
The role of ADAM proteases in ectodomain shedding. Membrane-bound ADAM recognizes specific cleavage site in the extracellular membrane-proximal region of membrane-tethered substrates, following which, it enzymatically cleaves substrates to facilitate their release as biologically active soluble ectodomains.

**Figure 4 cancers-11-01218-f004:**
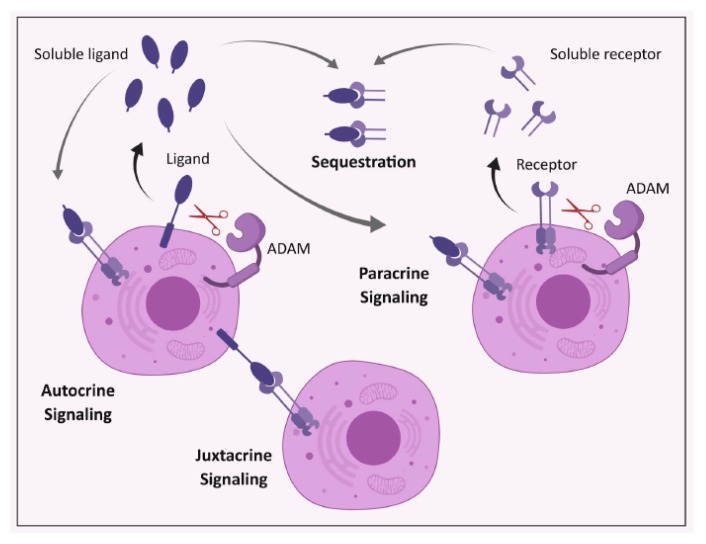
Modes of action of ectodomain proteins. Schematic representation of the various autocrine, paracrine and/or juxtacrine modes of action by which biologically active processed ADAM17 substrates (i.e., ectodomains) act on cells.

**Figure 5 cancers-11-01218-f005:**
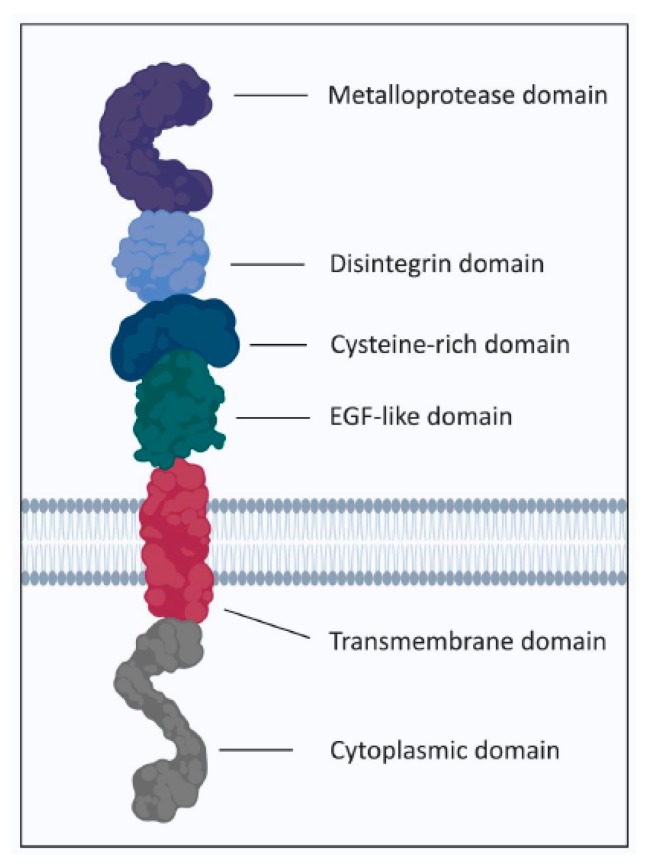
Overview of the domain structure of ADAM proteases.

**Figure 6 cancers-11-01218-f006:**
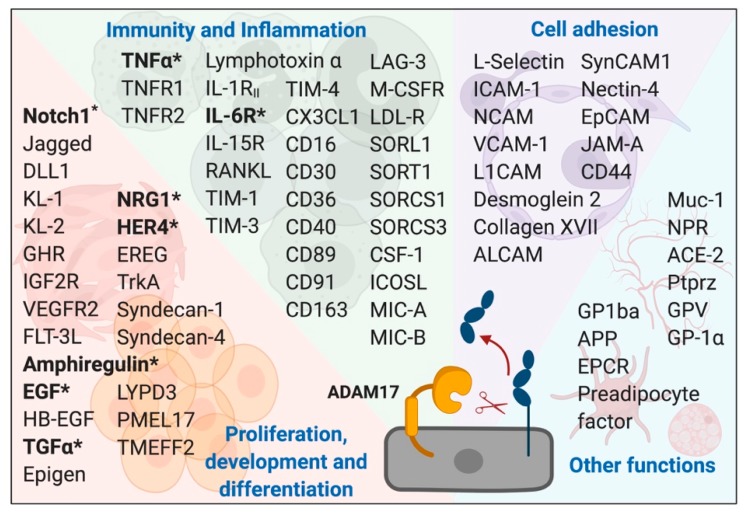
ADAM17 substrate repertoire. Shown are the >70 known ADAM17 substrates grouped according to their implicated role in numerous cellular processes (highlighted in bold blue font). ADAM17 substrates associated with lung cancer are denoted in bold font and by an asterisk.

**Figure 7 cancers-11-01218-f007:**
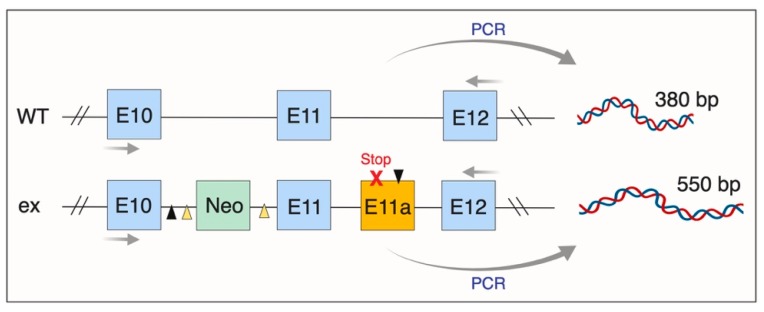
Overview of the genetic manipulation strategy used to create the *Adam17*^ex/ex^ mice. Shown are the exon regions for both wild-type (WT) and hypomorphic (ex) *Adam17* alleles. For generating the ex allele, the neomycin resistance gene in the targeting cassette is depicted “Neo”, along with the introduced stop codon in the artificial exon 11 (E11a). Arrows depict PCR primers use to amplify and distinguish specific regions of the WT and ex alleles.

**Figure 8 cancers-11-01218-f008:**
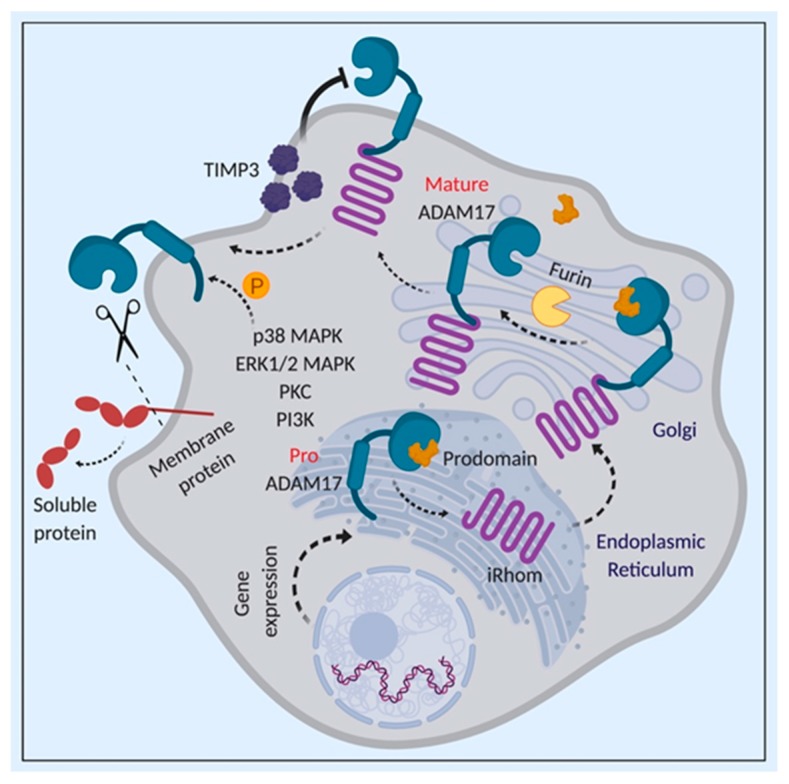
Overview of ADAM17 activity regulation. Schematic representation of the expression of immature proADAM17 polypeptide in the endoplasmic reticulum, following which it is transported to the golgi apparatus via iRhom-mediated mechanism. The proADAM17 then undergoes a maturation step, which requires its inhibitory N-terminal prodomain to be cleaved off by furin protease, following which, mature ADAM17 is transported to the cell membrane. The activity of ADAM17 can also be inhibited by TIMP3 or activated via phosphorylation by MAPKs, PI3K and PKC.

**Figure 9 cancers-11-01218-f009:**
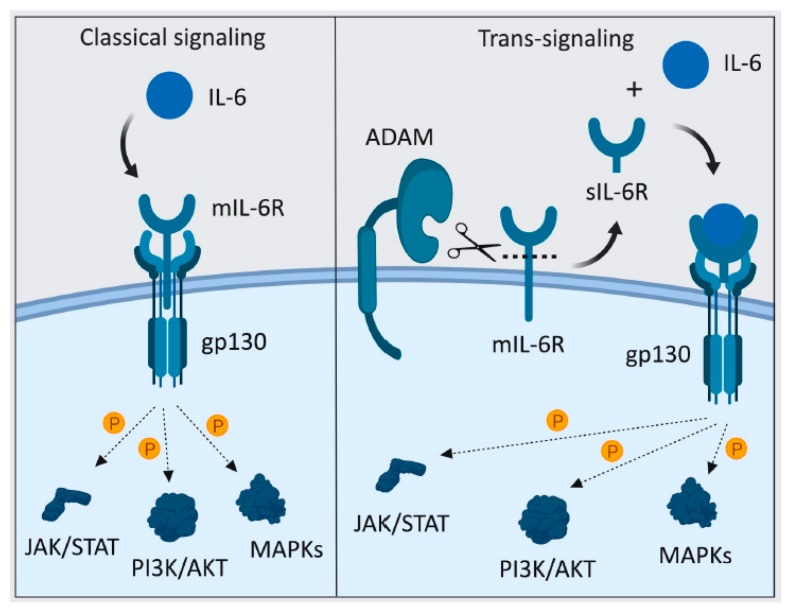
Classical IL-6 signaling via membrane-bound (m) IL-6R, and trans-signaling via soluble (s) IL-6R. Left panel: For classical signaling, upon IL-6 binding to its membrane-bound (m) IL-6R, a complex is formed with the signal-transducing gp130 co-receptor to elicit intracellular signaling via numerous signaling cascades. Right panel: For trans-signaling, ADAM proteases (e.g., ADAM17) cleave mIL-6R to release soluble (s) IL-6R, which then binds to free IL-6 to form a signal-transducing receptor complex with gp130. Since gp130 is ubiquitously expressed, IL-6 trans-signaling can both amplify the signal output on mIL-6R-expressing cells, as well as elicit IL-6 signaling in cells not expressing mIL-6R.

**Table 1 cancers-11-01218-t001:** Published clinical and preclinical evidence supporting the role of inflammation in NSCLC.

Factor	Main Effect	Model	Reference
Neutrophils	Poor prognosis	Human patients	[88]
Macrophages	Promoting lung carcinogenesis	Urethane-induced lung cancer mouse model	[89]
IL-17^+^ T cells	Enhancing tumour proliferation and angiogenesis	*Kras* mutant mouse model	[92]
Commensal microbiota	induce tumour cell proliferation via activating lung-resident γδ T cells	Mouse model harbouring *Kras* mutation and *Trp53* deficiency	[93]
IL-17	Promoting tumour angiogenesis	NSCLC xenografts	[94]
IL-6	Activation of IL-6 trans-signaling	*Kras* mutant mouse model	[95,96,97]
IL-8	Promoting tumour angiogenesis	Human patients	[98,99,100,101]
IL-1β	Enhance lung cancer incidence and mortality	Human patients (CANTOS)	[103]

**Table 2 cancers-11-01218-t002:** A summary of some therapeutic inhibitors of ADAM17.

Agent	Type of Agent	Disease Setting	Substrate Inhibited	Reference
INCB3619	Small molecule inhibitor	Breast cancer and NSCLC	Heregulin	[182]
INCB7839	Small molecule inhibitor	Breast cancer	HER2	[183,184]
KP-457	Small molecule inhibitor	Thrombus formation	Glycoprotein Ibα (GPIbα)	[185]
Prodomain	Peptide	LAC and inflammatory diseases	TNFα and IL-6R	[29,131,143]
D1(A12)	Antibody	Triple-negative breast cancer and head and neck squamous cell carcinoma	TNFα, TGFα, amphiregulin and TNFR1	[174,186]
A9(B8)	Antibody	Pancreatic ductal adenoma	Amphiregulin	[180]

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
