# Peer review of "ADAM17: An Emerging Therapeutic Target for Lung Cancer"

_cancers, 2019, doi:10.3390/cancers11091218_

Round 1
Reviewer 1 Report
Overall a very well written comprehensive review which I expect to be of great interest to people working in the field.
Just a few minor issues should be addressed before publication:
1. Part 2.1 Genetics
KRAS Mutations: In the paragraph about Kras-targeted therapies, a few sentences regarding recent efforts in the field of mutant KRAS-targeted vaccines, should be added. The same is true for the MEK/ERK targeting strategies, where the works regarding MEK + SHP2 inhibition in lung and pancreatic cancer published in 2018 are missing.
EGFR Mutations: In the last paragraph I would suggest moving the conclusion about monoclonal antibodies from line 203 to 198.
2. Inflammation
It would be good to summarize the information from this paragraph in a table. Listing the factor (neutrophils, Il-17 etc.), the main effect ( i.e. poor prognosis, promotion of tumor growth), the model and the respective reference.
3. ADAM17
The authors should consider adding a figure depicting the different strategies used to generate in vivo models of ADAM17 deficient mice, and the main findings derived from these models.
As well as a figure summarizing of the relationship between ADAM17 and EGFR ligands, ADAM17 and Notch Signaling as well as ADAM17 and sIL6-R and the potential benefits of ADAM17-targeted therapies.
Reviewer 2 Report
This review article by Saad et al. provides a comprehensive synthesis of pathways contributing to lung tumorigenicity and their relation to ADAM17 activity. Although there are several existing reviews on ADAM17 as a therapeutic target for NSCLC, the authors set themselves apart with a timely piece of writing that incorporates a number of recent articles illuminating ADAM17’s roles, especially in the context of KRAS- and EGFR-hyperactivated lung cancer. This work is not only informative but also accessible to the more general reader as ample background is provided. The review begins by running readers through an introduction to lung cancer and its various subtypes. It then demonstrates the need for more effective therapies, highlights the two most prevalent genetic drivers of lung adenocarcinoma - KRAS and EGFR - and their downstream pathways, as well as the contribution of tobacco smoke and inflammation to tumorigenesis and the underlying pathways involved. The authors then provide a compelling case for ADAM17 as a therapeutic target - covering early work revealing ADAM17’s involvement in cancer, as well as mechanistic details of ADAM17’s roles in promoting proliferation and tumor growth in LAC.
The authors have made effective use of figures, which are both well-designed and complement the text to aid reader understanding.
Some suggestions:
While the discussion on lung cell types and the cell-of-origin responsible for distinct lung cancer subtypes (Lines 54-69) is informative, it is not revisited in later sections and could be distracting for readers. Given that the focus of this review is on ADAM17, the authors could consider omitting this section, which would provide room for a deeper discussion of another more relevant aspect.
The scoping of the review article can be slightly confusing at times. The abstract begins by narrowing the focus to lung adenocarcinoma (LAC), yet the sections on KRAS and EGFR mutations were more general and referred to other cancers. The later section on inflammation referred broadly to non-small cell lung cancer (NSCLC), whereas discussion of the relevance of ADAM17 in lung cancer shifted the focus back to LAC. Better organization and signposting between sections could be considered, and would be useful in highlighting the transferability of insights gained from ADAM17 in LAC to other cancers.
The section on inflammation is detailed and informative, highlighting several aspects from microbiome involvement to the contribution of innate and adaptive immune cells. Perhaps the authors could consider the use of subheadings to provide readers with a more coherent structure for easier reading.
In section 3.1 on the requirement of ADAM17 for the shedding of various effector proteins, a figure or table could be used to convey selected information for key effectors to complement the text.
Section 3.2 on the regulation of ADAM17 may benefit from the inclusion of a schematic diagram.
ADAM17 was first mentioned quite early on (lines 139-143) within the KRAS mutations part in section 2.1, although ADAM17 is only properly introduced later in the review. This could be slightly harder to follow and requires close re-reading on the part of the reader. Perhaps the authors could consider re-organizing this or make a note preempting readers about the later sections.
Given that the authors have selected to include the therapeutic potential of inhibiting ADAM17 as a key focus of the review, it would be insightful to expand on current ADAM17 inhibition strategies. As there are several ADAM17-targeting molecules that are in pre-clinical trials, some reference could be provided, perhaps in table form, for readers who are interested in the progress and development of particular inhibitors.
The conclusion seemed a little abrupt. It would be nice to read more about the authors’ specific views on where future work in this field is headed.
Minor word choice suggestions that may help the article read better:
Line 36: 1.8 million lung cancer cases are diagnosed each year
Line 47: ‘cytotoxic’ and ‘chemotherapy’ are redundant: non-selective chemotherapy cocktail of...
Line 68: ‘driver’ instead of ‘diver’
Line 92: ‘Interestingly’ is a peculiar word choice as the nature of KRAS mutations resulting in constitutive activation of RAS proteins is already well-established
Lines 101-102: What is meant by the distinctive RAS pathway dependency and its implications could perhaps be elaborated upon
Line 118: Comma after ‘for instance’
Lines 119 to 122: Why is it paradoxical?
Line 123: and are often excluded
Lines 386-387: Some grammar inconsistencies here
Line 485: Role of ADAM17 in hair follicle differentiation is not central to this review’s focus on cancer/lung disease and could be omitted.
Lines 492-494: some grammar inconsistencies here
Reviewer 3 Report
Overall, the review is well written and informative. The one critical issue, however, is that the document does not focus on ADAM17 primarily. Fully half the review is a (informative) description of lung cancer biology with only weak linkage to the later ADAM17-specific discussion. ADAM17 does not receive significant notice till line 314, more than halfway into the manuscript. Furthermore, the linkage between ADAM17 and lung cancer therapy, while present, seems relatively weak, and it is unclear if it is really at the stage that a comprehensive review is warranted.
Furthermore, this manuscript is supposed to be part of a JAK/STAT special issue. While IL-6 (which induces JAK/STAT signaling) and it's linkage to ADAM17 is discussed, this is a minor part of the paper.
Thus, the overall critique of the manuscript is one of content emphasis. The information presented is of interest to read and comprehensive, but the focus on ADAM17 and JAK/STAT is lacking. For publication, I would advise further ADAM17 content and emphasis on JAK/STAT while trimming down the earlier sections on general lung cancer biology.
Author Response
Thank you for your recommendation to respond only to Reviewers 1 and 2, which therefore sufficiently addresses Reviewer 3's comments.
Round 2
Reviewer 3 Report
Dear Authors,
Thank you for revising the manuscript so thoroughly and so promptly. Your response to the feedback has been admirable. The additional figures and tables and reorganization greatly enhances the manuscript. The review is acceptable for publication.